# Symptom evolution following the emergence of maize streak virus

Adérito L Monjane[1,2†], Simon Dellicour[3,4†*], Penelope Hartnady[5], Kehinde A Oyeniran[5], Betty E Owor[6], Marion Bezuidenhout[7], Daphné Linderme[7], Rizwan A Syed[7], Lara Donaldson[7], Shane Murray[7], Edward P Rybicki[7], Anders Kvarnheden[2], Elham Yazdkhasti[2], Pierre Lefeuvre[8], Rémy Froissart[9], Philippe Roumagnac[10,11], Dionne N Shepherd[7,12], Gordon W Harkins[13], Marc A Suchard[14], Philippe Lemey[3], Arvind Varsani[15,16], Darren P Martin[5*]

[1]Fish Health Research Group, Norwegian Veterinary Institute, Oslo, Norway; [2]Department of Plant Biology, Swedish University of Agricultural Sciences, Uppsala, Sweden; [3]Department of Microbiology, Immunology and Transplantation, Rega Institute, Laboratory for Clinical and Epidemiological Virology, KU Leuven - University of Leuven, Leuven, Belgium; [4]Spatial Epidemiology Laboratory (SpELL), Université Libre de Bruxelles, Brussels, Belgium; [5]Computational Biology Division, Department of Integrative Biomedical Sciences, Institute of Infectious Diseases and Molecular Medicine, University of Cape Town, Observatory, Cape Town, South Africa; [6]Department of Agricultural Production, School of Agricultural Sciences, Makerere University, Kampala, Uganda; [7]Molecular and Cell Biology Department, University of Cape Town, Cape Town, South Africa; [8]CIRAD, UMR PVBMT, La Réunion, France; [9]University of Montpellier, Centre National de la Recherche Scientifique (CNRS), Institut de recherche pour le développement (IRD), UMR 5290, Maladie Infectieuses & Vecteurs: Écologie, Génétique Évolution & Contrôle" (MIVEGEC), Montpellier, France; [10]CIRAD, BGPI, Montpellier, France; [11]BGPI, INRA, CIRAD, SupAgro, Univ Montpellier, Montpellier, France; [12]Research Office, University of Cape Town, Cape Town, South Africa; [13]South African Medical Research Council Bioinformatics Unit, South African National Bioinformatics Institute, University of the Western Cape, Bellville, South Africa; [14]Department of Biomathematics, David Geffen School of Medicine, University of California, Los Angeles, Los Angeles, United States; [15]The Biodesign Center for Fundamental and Applied Microbiomics, Center for Evolution and Medicine, School of Life Sciences, Arizona State University, Tempe, United States; [16]Structural Biology Research Unit, Department of Integrative Biomedical Sciences, University of Cape Town, Cape Town, South Africa

*For correspondence:
Simon.Dellicour@ulb.ac.be (SD);
darrenpatrickmartin@gmail.com
(DPM)

†These authors contributed
equally to this work

Competing interests: The
authors declare that no
competing interests exist.

Reviewing editor: Antonis
Rokas, Vanderbilt University,
United States

**Abstract** For pathogens infecting single host species evolutionary trade-offs have previously been demonstrated between pathogen-induced mortality rates and transmission rates. It remains unclear, however, how such trade-offs impact sub-lethal pathogen-inflicted damage, and whether these trade-offs even occur in broad host-range pathogens. Here, we examine changes over the past 110 years in symptoms induced in maize by the broad host-range pathogen, maize streak virus (MSV). Specifically, we use the quantified symptom intensities of cloned MSV isolates in differentially resistant maize genotypes to phylogenetically infer ancestral symptom intensities and check for phylogenetic signal associated with these symptom intensities. We show that whereas symptoms reflecting harm to the host have remained constant or decreased, there has been an increase in how extensively MSV colonizes the cells upon which transmission vectors feed. This

demonstrates an evolutionary trade-off between amounts of pathogen-inflicted harm and how effectively viruses position themselves within plants to enable onward transmission.

## Introduction

Maize was first introduced to Africa from South America in the early 1500s, and within 300 years became one of the most important food crops on the continent (*McCann, 2001*). The rapid intensification of maize cultivation in the 1800s coincided with the emergence of maize streak disease in southern Africa (*Fuller, 1901*; *Harkins et al., 2009*); a disease that has subsequently spread to all maize growing parts of Africa and today persists as one of the most serious threats to the food security of sub-Saharan Africans (*Bosque-Pérez, 2000*; *Martin and Shepherd, 2009*). The causal agent, a maize-adapted strain of maize streak virus (MSV; genus *Mastrevirus*, family *Geminiviridae*), is apparently a recombinant of two wild-grass adapted MSV strains: MSV-B and MSV-F/G (*Varsani et al., 2008*). Today the geographical range of this 'MSV-A' strain (*Kraberger et al., 2017*) coincides with the combined ranges of its various insect transmission vector species, all of which are leafhoppers in the genus *Cicadulina*, and includes most of Africa and various Atlantic and Indian Ocean islands (*Bosque-Pérez, 2000*). Crucially, the host-range of MSV-A is also broadly coincident with that of its vector species, and includes more than 100 different grass species (*Damsteegt, 1983*; *Kraberger et al., 2017*).

The rise of MSV-A is a typical viral emergence story, and as such, efforts to reconstruct what happened after MSV-A first began infecting maize could potentially yield generalisable insights into how pathogens evolve following successful host-range changes. It would be particularly informative to determine how the symptoms that MSV-A induces in maize have evolved since the virus first emerged in the mid to late 1800s. Although it is expected that pathogens like MSV-A will evolve over time to maximise their probability of transmission, there is a spectrum of possible ways in which this transmission optimisation imperative might impact the evolution of pathogen-mediated molecular processes that underlie infection symptoms (*Bull and Lauring, 2014*; *Cressler et al., 2016*; *Doumayrou et al., 2013*; *Escriu et al., 2003*; *Mauck et al., 2010*; *Read, 1994*). Various theories, focusing primarily on trade-offs between host mortality rates, infection durations and pathogen transmission rates, predict that under natural selection, transmission probabilities and hence the basic reproductive number of a pathogen, should be optimised at intermediate rates of pathogen-induced host mortality (*Alizon et al., 2009*; *Anderson and May, 1982*).

Because of their general disregard for sub-lethal amounts of host harm, it is unclear how relevant such theories might be for explaining the evolution of MSV-induced disease symptoms. MSV and many other plant viruses only rarely kill their hosts (i.e. pathogen attributed mortality rates are generally close to zero) and infections generally persist for the remainder of a host's life. Also, selection pressures acting on crop-infecting viruses such as MSV will not necessarily yield the same outcomes as for pathogens that infect humans, other animals or uncultivated plants because crop plants frequently have either very short natural lifespans or are destructively harvested long before their natural lifespans expire. Regardless of how these factors might influence the evolution of symptoms over time, it is expected that infection symptoms that are associated with decreased transmission probabilities - perhaps high host mortality rates or reduced host reproduction rates - should evolve to either remain mild, or to become less severe. Conversely, symptoms that are associated with increased transmission probabilities - such as increased lesion sizes or lesion colours that are more attractive to insect vector species - should evolve to become more extreme. Therefore, all that would be required to detect relationships between transmission probabilities and the intensity of virus-induced disease symptoms would be to first identify quantifiable symptom types that might impact transmission, and then to determine whether these vary in some concerted way during the adaptation of a virus such as MSV to a new host species such as maize.

The most obvious symptom of a MSV-A infection that is likely to impact transmission is the occurrence of distinct yellow streaks centred along the primary, secondary and tertiary veins of maize leaves. These streaks arise as a consequence of impaired chloroplast formation within MSV-infected photosynthesing cells that surround veins (*Engelbrecht, 1982*; *Pinner et al., 1993*). The proportion of a leaf's surface that is covered in streaks reflects, at least in part, the degree to which viruses originating in the vasculature manage to infect surrounding photosynthesing tissues during the early

developmental stages of the leaf's formation (*Lucy, 1996*). The intensity of chlorosis within streaks varies both between MSV variants infecting the same host, and between different host genotypes that are infected by the same MSV variant (*Martin et al., 1999*; *Pinner et al., 1993*; *Schnippenkoetter et al., 2001*). Depending on the number of intact chloroplasts within MSV-infected photosynthesising cells, streaks can range in colour from pale green through yellow to white (*Engelbrecht, 1982*; *Pinner et al., 1993*). Therefore, whereas the spatial arrangements and dimensions of chlorotic lesions on MSV-infected maize leaves directly indicate the locations and numbers of leaf cells that are infected by MSV (*Lucy, 1996*), the colour of chlorotic streaks indicates the degree to which photosynthesis is impaired within infected cells (*Pinner et al., 1993*).

Another easily quantifiable symptom of MSV infections that may impact the probability of virus transmission is leaf stunting. While leaf stunting is often attributed to reduced carbohydrate production due to virus-mediated impairment of photosynthesis (*Shepherd et al., 2010*), it is also probably a consequence of MSV directly interfering with the cell cycle progression control mechanisms of infected cells (*Lucy, 1996*; *McGivern et al., 2005*; *Shepherd et al., 2005*). MSV-induced cell cycle abnormalities could also account for other less commonly described bi-lateral leaf asymmetry symptoms. These symptoms can include off-centre leaf midribs, bending or twisting of leaves where one half of a leaf is more stunted than the other, and curling or corrugation of leaves caused by varying degrees of veinal swelling along the leaf-length.

Whereas measurements of leaf stunting or leaf deformation might be used as indicators of MSV-mediated harm to host plants, measurements of chlorotic areas - the proportion of leaf surface areas that are covered by streaks - might be used as indicators of how effectively MSV colonizes host cells. Since MSV can only be acquired by its insect transmission vectors when they feed directly on chlorotic lesions (*Peterschmitt et al., 1992*), and no report exists of any successful MSV transmissions from MSV-infected plants that do not display streak symptoms, chlorotic leaf areas are likely correlated to some degree with transmission probabilities. Further, measurements of streak colours could be indicators of both harm to the host, and the transmission fitness of MSV. Specifically, whereas the intensity of chlorosis within streaks reflects the extent of chloroplast destruction within infected photosynthesising cells, it could also be associated with the probability of onward MSV transmission. This is because the sap-feeding insects that transmit viruses such as MSV tend to prefer pale-yellow over green when selecting feeding sites on leaves (*Fereres and Moreno, 2009*; *Hodge and Powell, 2008*; *Isaacs et al., 1999*).

Irrespective of the precise underlying genetic or physiological causes of MSV-induced disease symptoms, any associations that exist between the fitness of MSV and the intensity of the infection symptoms that it causes, should be detectable by quantitative analyses of different symptom types that are produced in maize by an assortment of MSV isolates sampled over the ~110 years following the emergence of the virus as a maize pathogen.

Here we undertake such an analysis. Since there are no available MSV samples from before 1979, we use the quantified chlorotic areas, intensities of chlorosis, leaf deformation and leaf stunting symptoms caused by 59 diverse field isolates of MSV-A sampled between 1979 and 2007 to phylogenetically infer the evolution of symptoms produced by MSV-A in maize between ~1900 and 2007. Finally, to verify the accuracy of these symptom inferences we infer the genome sequences of seven ancestral MSV-A variants, synthesise infectious clones of these in vitro and directly compare actual and inferred symptom intensities. What we find is that patterns of MSV-A symptom evolution over the last century are broadly consistent with there being a trade-off between transmission probabilities and the amount of harm MSV inflicts on infected maize.

## Results and discussion

### Increased within host virus spread is not necessarily associated with harm to the host

To determine the degrees to which symptoms of MSV-A infections have evolved since the time of the most recent common ancestor (MRCA) of all known MSV-A isolates, we quantified four infection symptom types (chlorotic areas, intensities of chlorosis, leaf deformation and leaf stunting) for 59 MSV-A isolates and one MSV-B isolate in three differentially MSV-resistant maize genotypes. These were Golden Bantam (Sakata, South Africa) which is very susceptible to infection by MSV, displays

severe MSV infection symptoms and will hereafter be referred to as the 'S' (for susceptible) maize genotype; STAR174 (Starke Ayres, South Africa) which has a moderate degree of resistance to MSV, displays moderate MSV infection symptoms and will hereafter be referred to as the 'M' (for moderate) maize genotype, and PAN77 (Pannar Seed Company, South Africa) which has a high degree of resistance to MSV, displays mild MSV infection symptoms, and will hereafter be referred to as the 'R' (for resistant) maize genotype.

We first attempted to determine whether evidence existed of correlations between the measured symptom types (*Figure 1*). It was apparent that in the S maize, degrees of leaf stunting caused by MSV were positively correlated with chlorotic areas and leaf deformation. In the R maize there was also a strong positive correlation between leaf stunting and chlorotic areas, but there was a negative correlation between leaf stunting and the intensity of chlorosis. Therefore, for both the S and R maize the higher the proportion of photosynthesising leaf cells that are infected by MSV, the greater the amount of leaf stunting that occurs. Chlorotic areas were negatively correlated with the intensity of streak chlorosis in the M and R maize (*Figure 1*). This indicates that as greater numbers of photosynthesising cells in the leaves of the more MSV resistant maize genotypes become infected, the amount of chloroplast destruction within the infected cells tends to decrease.

Collectively, these results indicate that symptoms which are indicative of harm to the host - such as the intensity of chlorosis, leaf deformation and leaf stunting - are not, as one might expect, always positively correlated with chlorotic areas; a symptom type indicative of how successfully viruses have colonized photosynthesising cell populations within maize leaves (*Lucy, 1996*). It is apparent, therefore, that harm to the host could, to some degree at least, be mitigated without strongly impacting the capacity of MSV to infect and replicate within host cells. Put simply, selection acting on particular symptom types will not necessarily impact how the other symptom types evolve.

## MSV symptom measurements contain detectable phylogenetic signals

All of the symptom measurements for the 59 phenotyped MSV-A isolates contained detectable phylogenetic signals, in that sequences that were associated with similar symptom intensities tended to cluster together to some degree within the MSV-A phylogenetic tree (*Figures 2* and *3*, see also *Figure 2—figure supplements 1* and *2*). Specifically, Pagel's $\lambda$ values - where 0 = no correlation and 1 = perfect correlation between symptom measurements and phylogenetic placement - for every combination of symptom and host type ranged between 0.251 for leaf deformation in the M maize genotype, and 0.706 for intensity of chlorosis in the M maize genotype, with the 95% credibility intervals of these values excluding zero for every symptom x host comparison (*Figures 2* and *3*).

This association between the symptoms induced by MSV isolates and their phylogenetic placement indicated that, just as it is possible to infer ancestral genome sequences based on the phylogenetic relationships of an observed sample of genome sequences, it should be possible to meaningfully infer the symptom intensities that were induced by ancestral MSV variants based on the observed symptom intensities of the 59 phenotyped MSV-A isolates.

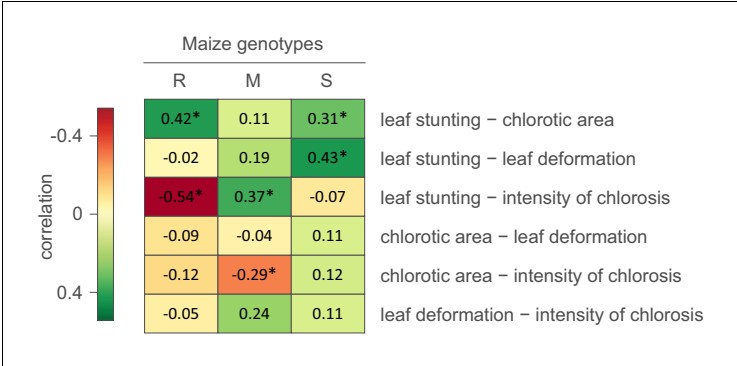

**Figure 1.** Correlations between MSV-A symptom measurements. For each specific maize genotype and pair of traits, correlation estimates accounting for the phylogenetic relatedness of viral isolates are indicated by the colour gradient. A correlation estimate was considered significant (indicated by an asterisk) if the 95% HPD interval of the posterior distribution of the estimate excluded zero.

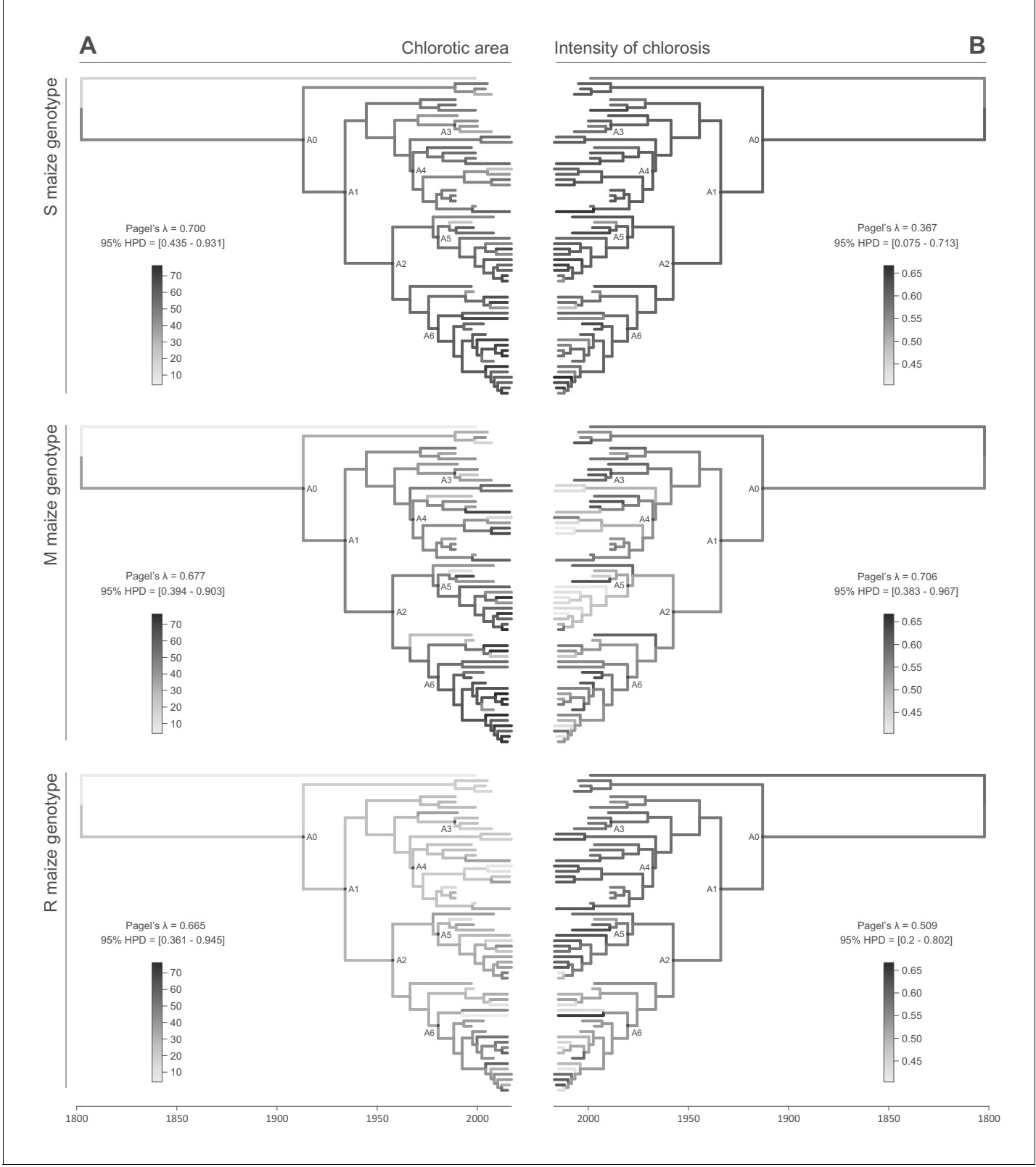

**Figure 2.** Continuous character mapping of symptom intensities, chlorotic area (A) and intensity of chlorosis (B), onto the MSV-A maximum clade credibility tree. Symptom intensities of ancestral viruses were phylogenetically inferred based on those observed in sensitive (S), moderately resistant (M) and resistant (R) maize genotypes infected by 59 MSV-A isolates and one MSV-B isolate (the outgroup), sampled between 1979 and 2007. The intensities of symptoms are represented by coloured branches on a grey-scale. 'A0-A6' indicate the ancestral nodes of the constrained clades for which

*Figure 2 continued on next page*

*Figure 2 continued*

MRCA sequences were inferred, synthesised and used to verify symptom intensity estimates. For each maximum clade credibility tree, we also report Pagel's λ values (*Pagel, 1999*), which indicate the degree of phylogenetic signal associated with the various symptom measurements. A version of the tree with taxon labels is given in *Figure 2—figure supplement 1*.

The online version of this article includes the following figure supplement(s) for figure 2:

**Figure supplement 1.** Maximum clade credibility tree (from *Figure 2*) with taxon labels and posterior probabilities displayed at internal nodes.

**Figure supplement 2.** Maximum clade credibility tree (from *Figure 2*) with taxon labels and 95% highest probability density (HPD) intervals reflecting the uncertainty of inferred node ages (green bars).

**Figure supplement 3.** Changes in MSV-A symptom intensities over time across viral lineages descending from the most recent common ancestor of all known MSV-A isolates.

**Figure supplement 4.** Maximum likelihood phylogenetic tree indicating the evolutionary relationships between the 59 MSV-A isolates selected in this study and all other available MSV-A isolates.

## Increase of MSV-induced chlorotic leaf areas over the past ~110 years of MSV evolution

We used the continuous diffusion model implemented in BEAST 1.10 (*Suchard et al., 2018*) to infer the intensities of infection symptoms produced by ancestral MSV-A sequences since the emergence of MSV-A as a serious maize pathogen in the late 1800s. To identify general trends in symptom intensity changes across all of the sampled MSV-A lineages, we mapped symptom intensities to the time-scaled MSV phylogeny (i.e. the maximum clade consensus tree; *Figures 2* and *3*) and then examined the averaged symptom intensities associated with phylogenetic tree branches through time (*Figure 4*). This revealed that since 1979, the time of the earliest sampled MSV-A isolates investigated here, MSV-A lineages have been progressively inducing, on average: (1) higher percentage chlorotic areas in all tested hosts, but most notably in the S and M maize genotypes; (2) lower intensities of streak chlorosis in the M and R maize genotypes; (3) more leaf deformation in the S maize genotype; and (4) slightly lower degrees of leaf stunting in the S and M maize genotypes (*Figure 4*).

Of all of the analysed symptom types, only average chlorotic areas on all three maize genotypes (*Figure 4A*), and leaf stunting in the R genotype (*Figure 4D*), were predicted to have progressively increased between 1900 and 1979. It is important to stress, however, that the credibility intervals of these average symptom intensities are particularly broad for the 'pre-sampling' time-period between 1900 and 1979. This is because: (1) the uncertainty associated with predicted ancestral virus symptom intensity estimates is expected to increase as we extrapolate backwards in time before the earliest sampling dates; and (2), at any given time-point, different lineages were not all predicted to induce similar symptom intensities such that the averaged symptom intensities across these lineages are associated with an additional among-lineage degree of variance.

When considering the predicted intensity of the various symptom types along specific virus lineages, rather than averaging these across the entire sample of viruses, it is apparent that changes in symptom intensity were not predicted to have all occurred concomitantly across all MSV-A lineages. For example, when the predicted chlorotic area and leaf deformation estimates are overlaid on the MSV-A phylogeny (*Figures 2A* and *3A*, respectively), it is apparent that multiple independent large increases in leaf deformation and decreases in chlorotic area are predicted to have occurred across various terminal branches of the phylogeny (*Figures 2* and *3*). The sampled viruses at the tips of these branches tend to display large increases in leaf deformation or decreases in induced chlorotic areas relative to closely related viruses, frequently in two or three of the maize genotypes – and are dispersed throughout the phylogeny (i.e. they are not all close relatives to each other). The fact that there is no clustering within the phylogeny of viruses that display either large decreases in chlorotic area or large increases in leaf deformation symptoms compared to their nearest relatives, suggests that both increased leaf deformation and decreased chlorotic area phenotypes are maladaptive.

Conversely, the phylogenetic clustering of viruses that all share similar increases or decreases in the intensity of particular symptoms relative to the other sampled viruses would be consistent with ancestral sequences having undergone adaptive changes in the intensity of the symptoms that they induce. There are multiple phylogenetic clusters containing viruses displaying decreased intensity of chlorosis, and/or increased chlorotic area phenotypes relative to those predicted for ancestral viruses (*Figures 2* and *3*): this is an indicator that both decreased intensities of chlorosis, and increased chlorotic areas might be adaptive. It is noteworthy, however, that some of these clusters

of viruses that induce low intensities of chlorosis and/or high chlorotic areas, are not consistent between the three different maize hosts. This again suggests that such adaptations might, to some degree at least, be host-genotype specific.

Although there was an overall positive correlation between the chlorotic area and leaf stunting symptoms of the 59 phenotyped MSV isolates (especially in the S and R maize genotypes; *Figure 1*), there were no clear instances where both chlorotic areas and leaf stunting were predicted to have concomitantly increased in extent across individual internal branches of the MSV phylogeny. There were, however, a number of instances across terminal branches of the phylogeny where predicted decreases in chlorotic area were associated with concomitant predicted decreases in leaf stunting. The viruses at the tips of these branches induce lower chlorotic areas and/or less leaf stunting than their nearest relatives in two or three of the tested maize genotypes (*Figure 3B*).

## Potential impact of isolate sampling

It should be stressed here that each of the 59 phenotyped MSV-A isolates was a single clone that was sampled at random from a genetically heterogeneous population of viruses infecting an individual maize plant. It is well known that up to 10% of MSV genomes that are cloned from an infected plant will induce significantly reduced chlorotic areas and leaf stunting relative to other clones from the same infection that might better represent the dominant virus population within that plant (*Boulton et al., 1991a*; *Boulton et al., 1991b*). It is plausible, therefore, that the overall correlation between chlorotic areas and leaf stunting observed for the 59 phenotyped virus clones (*Figure 1*) is not solely a consequence of adaptive increases in induced chlorotic areas being causally linked with increased leaf stunting. Rather, the detected positive correlation between chlorotic areas and leaf stunting is probably attributable, at least in part, to the decreased chlorotic areas and leaf stunting symptoms produced by the approximately 10% (i.e. ~6/59) of examined MSV-A isolates which are expected to be slightly defective. Even if only affecting approximately 10% of the isolates, we acknowledge that this artefactual correlation could also, to some extent, impact the continuous character mapping of symptom intensities (*Figures 2* and *3*) and inferred changes in overall symptom intensities (*Figure 4*). However, because these 10% of isolates are a priori randomly distributed in the tree, their inclusion is mainly expected to add some random noise in the comparative analysis. Therefore, we do not expect a systematic bias in generating the trends we estimate and compare.

## Observed ancestral virus symptoms confirm symptom inferences

We selected seven nodes within the MSV-A tree and inferred the genome sequences of the ancestral viruses that are represented by these nodes (*Figures 2* and *3*; nodes labelled 'A0' to 'A6'). We then chemically synthesised these genome sequences, and directly measured the symptom phenotypes that they induced in the S, M and R maize genotypes (see the Materials and methods for further details).

We compared the observed symptoms produced by the synthesised ancestral viruses against those inferred by the Bayesian continuous trait mapping approach for these same viruses. This revealed that 16/21 of the chlorotic area, 14/21 of the intensity of chlorosis, 11/21 of the leaf deformation, and 16/21 of the leaf stunting measurements of the synthesised ancestral viruses were within the 95% confidence intervals of those inferred based on the phenotypes of contemporary MSV sequences (*Figure 5*). However, when comparing measured and inferred symptom intensities within each genotype separately, we did not find any significant correlation (determination coefficients of linear regressions all associated with a p-value>0.05). This is unsurprising given (i) only seven viruses were examined in this way and (ii) the high variance associated with both the inferred and observed estimates of symptom intensities. Observed symptoms deviated most notably from inferred symptoms for the R maize genotype, with 12/27 of the discordant symptom inferences being obtained for this host. In particular, 6/7 of the leaf deformation inferences for the R maize genotype were outside of the observed 95% confidence intervals of the measured estimates (*Figure 5C*).

Overall, the best correlation between the observed and inferred symptom measurements were obtained for the leaf stunting (Pearson $R^2$ = 0.74; slope = 0.87) and chlorotic area (Pearson $R^2$ = 0.58; slope = 0.84) measurements, indicating good agreement between the inferred and observed measurements of these symptom types in the three different maize genotypes. The agreement between the inferred and observed leaf deformation and intensity of chlorosis measurements

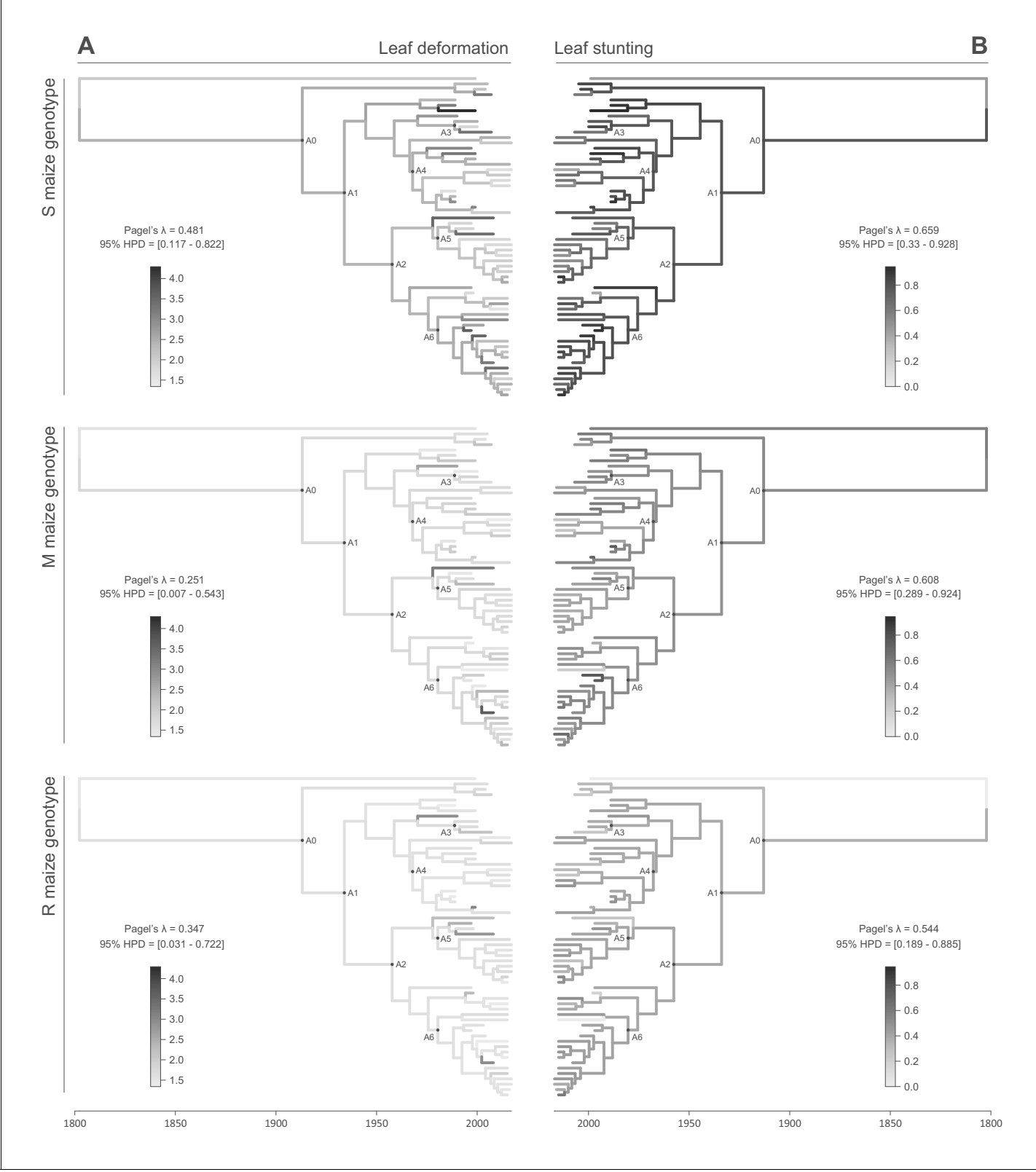

**Figure 3.** Continuous character mapping of symptom intensities, leaf deformation (**A**) and leaf stunting (**B**), onto the MSV-A maximum clade credibility tree. Symptom intensities of ancestral viruses were phylogenetically inferred based on those observed in sensitive (S), moderately resistant (M) and resistant (R) maize genotypes infected by 59 MSV-A isolates and one MSV-B isolate (the outgroup), sampled between 1979 and 2007. The intensities of symptoms are represented by coloured branches on a grey-scale. 'A0-A6' indicate the ancestral nodes of the constrained clades for which MRCA

*Figure 3 continued on next page*

*Figure 3 continued*

sequences were inferred, synthesised and used to verify symptom intensity estimates. For each maximum clade credibility tree, we also report Pagel's λ values (*Pagel, 1999*), which indicate the degree of phylogenetic signal associated with the various symptom measurements.

was less convincing (*Figure 5B* and *Figure 5C*). The greater degree of correlation between the observed and inferred chlorotic area and leaf stunting measurements is probably a consequence of these measurements containing more phylogenetic information than the others. For example, whereas the leaf stunting and chlorotic area measurements yielded the highest average Pagel's λ values, the leaf deformation measurements yielded both the lowest λ values and the lowest degree of correlation between observed and inferred symptom measurements (*Supplementary file 1*).

It should be stressed that disagreement between the observed and inferred symptom intensities for the synthesised ancestral viruses does not imply that the inferred estimates are incorrect. Ancestral sequences cannot usually be inferred with perfect accuracy: see *Supplementary file 2* for a list of genome sites in the inferred ancestral genomes that could not be resolved with a high degree of confidence. It is therefore very unlikely than any of the inferred ancestral genomes that we have tested actually ever existed. The synthesised viruses are, at best, close approximations of actual ancestral viruses.

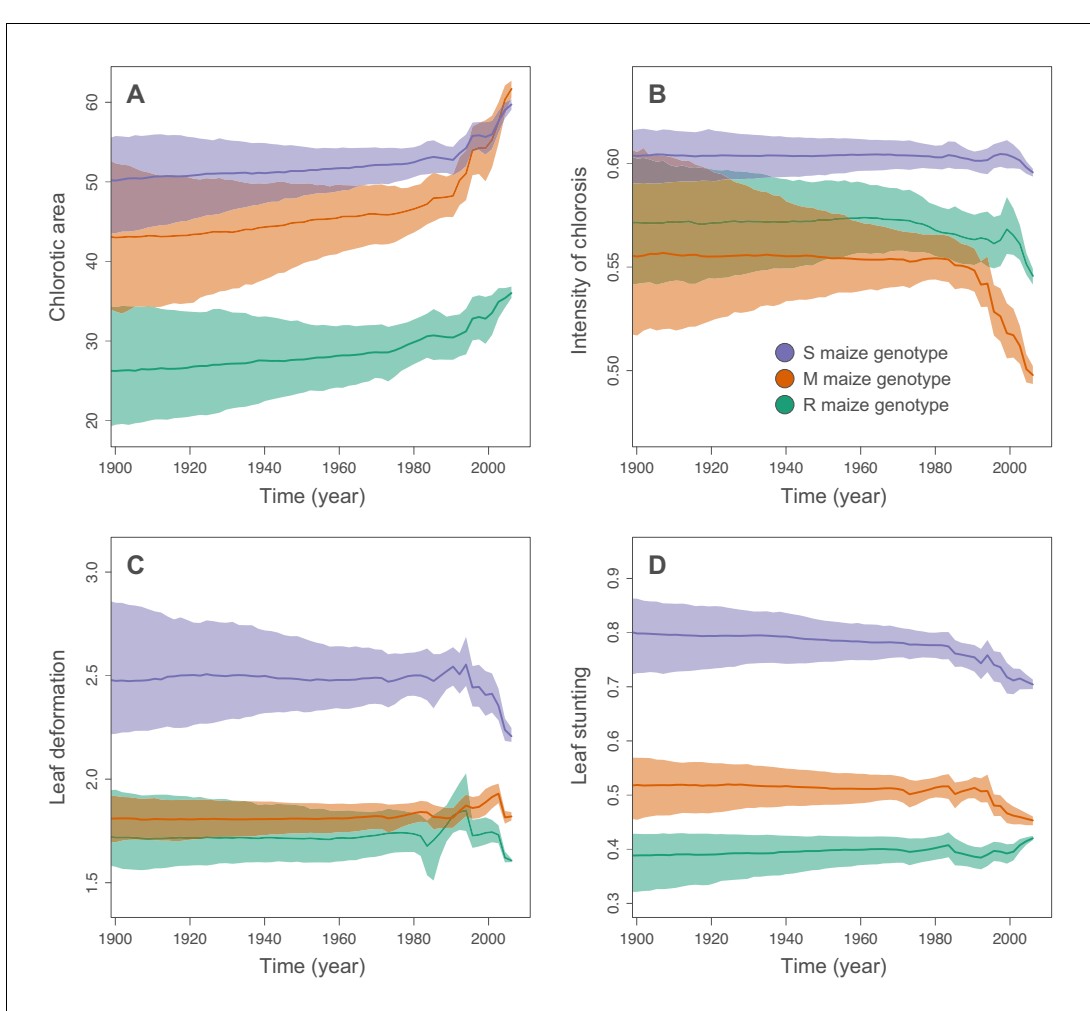

**Figure 4.** Inferred changes in MSV-A symptom intensities since 1900. The solid lines indicate the median values across the tree through time, and the shaded polygons represent the 80% highest posterior density (HPD) credible intervals: purple for the sensitive maize genotype (S), orange for the moderately resistant maize genotype (M) and green for the resistant maize genotype (R). Symptom intensities were here obtained from the phylogenetic analysis performed without including the MSV-B outgroup.

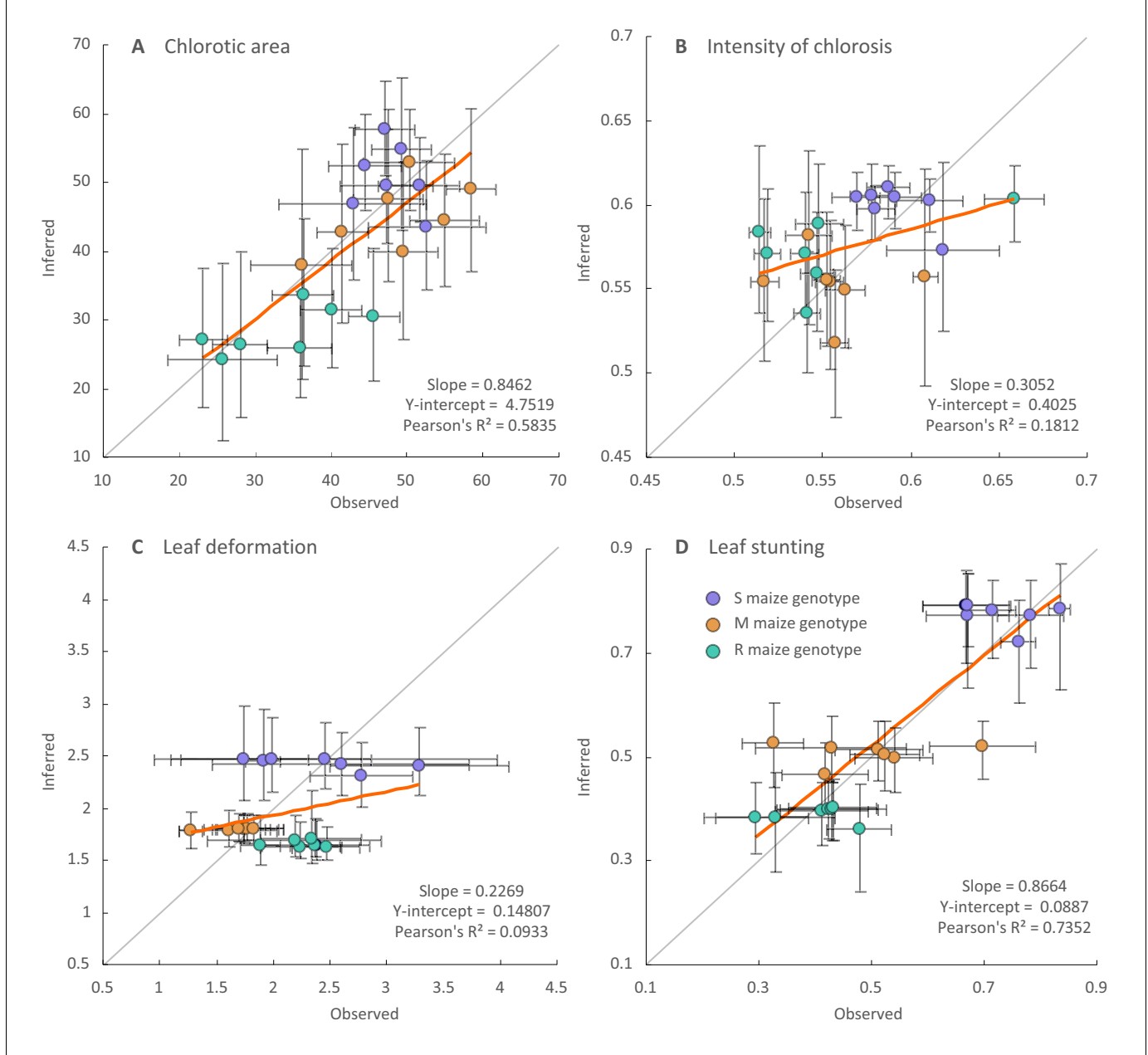

**Figure 5.** Regression analysis of the inferred and observed symptom intensities of synthesised ancestral MSV-A variants. Dots are coloured according to the maize genotype in which symptom intensities were inferred: purple for the sensitive maize genotype (S), orange for the moderately resistant maize genotype (M) and green for the resistant maize genotype (R). Vertical error bars reflect 95% credibility intervals of symptom intensity inferences for ancestral MSV-A variants, horizontal error bars represent 95% confidence intervals of the mean of observed symptom intensity measurements for synthesised versions of the ancestral MSV variants. The grey line has a slope = 1.0 and a y-intercept = 0: whereas points centred on this line would represent perfectly accurate inferences (i.e. instances where inferred symptom intensities = measured symptom intensities), those with associated error bars that intersect this line have credibility/confidence intervals that include coordinates where the inferred and observed symptom intensities are equal. Fitted regression lines are given in orange with slope parameters, Y-intercept parameters and Pearson's $R^2$ values given on the bottom right of each graph.

Nevertheless, the relative chlorotic areas and intensities of chlorosis induced by these synthetic approximations of actual ancestral viruses are broadly consistent with the computational inference that MSV-A is evolving towards the induction of increased chlorotic areas and decreased intensities of chlorosis (*Figure 2—figure supplement 3*). For example, whereas the 'oldest' of the synthesised ancestral viruses, A0 (which is estimated to have existed in ~1904), induced either the lowest or

second lowest chlorotic areas of all the synthesised ancestral viruses on all of the maize genotypes tested (*Figure 2—figure supplement 3A*), it also induced the highest intensities of chlorosis in all of these genotypes (*Figure 2—figure supplement 3B*).

The degrees of leaf deformation and leaf stunting induced by the synthesised ancestral viruses on the three maize genotypes also indicated possible trends in the evolution of these symptom types that were not revealed by the Bayesian continuous trait mapping approach. Specifically, whereas successively more recent MSV variants have tended to induce higher degrees of leaf deformation in the S and M maize genotypes (*Figure 2—figure supplement 3C*), these variants have tended to induce lower degrees of leaf stunting in the S maize genotype and higher degrees of leaf stunting in the M genotype (*Figure 2—figure supplement 3D*).

## MSV-A is apparently evolving towards a maximization of host colonization with a minimization of host harm

If the maize genotypes analysed here are in any way representative of those cultivated throughout Africa since 1900, these results suggest that MSV-A has over the past century been evolving towards the induction in most maize genotypes of increased chlorotic leaf areas and decreased intensities of chlorosis. Although in some maize genotypes this evolution has possibly been associated with increased degrees of leaf deformation and/or stunting, in others this does not seem to have been be the case.

Since MSV can only be acquired by its insect transmission vectors when they feed directly on chlorotic lesions (*Peterschmitt et al., 1992*), it is very likely that, by increasing the numbers of photosynthesizing maize leaf cells that it infects (i.e. by increasing induced chlorotic areas), MSV-A has increased the probability of its acquisition and onward transmission by insects.

The reason that decreased intensities of chlorosis might have been favoured by selection is not as obvious. Although it is plausible that selection has favoured an intensity of chlorosis that is maximally attractive to transmission vectors, it is similarly plausible that selection has also/instead favoured less chloroplast destruction so as to foster sufficient carbohydrate synthesis to prevent excessive host stunting. In this regard, further insect behavioural studies will be required to determine the relationships between the intensity of chlorosis, host stunting and transmission rates.

In any event, the changes in symptom intensities over time that we have both inferred and observed for MSV-A in maize are consistent with the hypothesis that the maximization of transmission probabilities which characterizes the adaptation of pathogens to their hosts (*Alizon et al., 2009*; *Anderson and May, 1982*), involves optimizing the balance between the amount of harm inflicted on the host and the extent to which pathogens colonize the host cell populations from which they will be transmitted to new hosts. Since MSV-A first arose in the mid- to late 1800s, it appears that the virus has achieved this balance by increasing the proportions of photosynthesising leaf cells that it infects, while concomitantly reducing the amount of chloroplast destruction that occurs within these cells.

The trade-off we identify is significant because it implies that even when pathogens have host ranges that include over 100 species, within the context of individual host species, they may still conform to evolutionary imperatives that are similar to those of narrow host-range pathogens. Possibly more significant, however, is that a trade-off between sub-lethal symptom severities and transmission rates in MSV alludes to a more general form of the popular transmission-virulence trade-off hypothesis (*Alizon et al., 2009*; *Anderson and May, 1982*) which has, until now, focused almost exclusively on the relationship between pathogen-induced host mortality and pathogen transmission rates.

## Materials and methods

### Virus symptom quantification

A set of 59 MSV field isolates (*Supplementary file 3*), representing the breadth of known diversity within the MSV-A strain, were used for symptom evaluation (*Figure 2—figure supplement 4*). These isolates were chosen from a collection of 704 fully sequenced MSV-A cloned genomes that were available in the University of Cape Town's MSV genome bank in December 2010. Their selection was based on sampling dates to ensure the broadest possible range of sampling times - from 1979 through to 2007.

Specifically, isolates were selected as follows: (i) we made and tested agroinfectious constructs for every virus that was sampled before 1990 and for which we had a molecular clone sampled directly from a source plant, (ii) we obtained and tested all agroinfectious clones that had previously been constructed in the 1990s and 1980s, and (iii) for viruses sampled between 2000 and 2007, we chose isolates that were representative of the major phylogenetic clades of viruses in eastern, western and southern Africa (*Figure 2—figure supplement 4*). Infectious clones of eight of the 59 selected isolates have been previously described (*Boulton et al., 1989*; *Martin et al., 2001*; *Peterschmitt et al., 1996*). Infectious clones of the remaining 51 isolates were constructed for this work as described by *Martin et al. (2001)*. Briefly, this involved taking linearised MSV genomes cloned in an *E. coli* plasmid vector, producing head-to-tail tandem copies of these genomes within their respective vectors using a series of partial restriction enzyme digests, transferring these genome 'dimers' into an *E. coli*/*Rhizobium radiobacter* (updated species name for *Agrobacterium tumefaciens*) binary plasmid vector (pBI121) and then transferring these recombinant plasmids, referred to as infectious constructs, into *R. radiobacter.*

Symptoms produced by each of the 59 MSV-A isolates and one MSV-B isolate (AF239960; MSV-B [ZA_VW_1989]) - a closely-related grass-adapted MSV strain (*Willment et al., 2002*) - were examined in three different maize genotypes that are known to vary in their degree of MSV resistance: (i) Golden Bantam that is very susceptible to infection by MSV ('S' genotype), (ii) STAR174 that has a moderate degree of resistance to MSV ('M' genotype), and (iii) PAN77 that has a high degree of resistance to MSV ('R' genotype). Whereas the R genotype is a white maize hybrid containing the MSV-1 resistance gene that has been deployed throughout sub-Saharan Africa within improved 'MSV-resistant' maize varieties since the 1980 s (*Bosque-Pérez, 2000*), the S and M maize genotypes were chosen as representatives of less MSV resistant maize varieties. The degrees of MSV resistance displayed by these three maize genotypes span those observed for commercial cultivars that have been widely used throughout Africa since at least the 1980s (*Martin et al., 1999*). Crucially, despite the widespread availability since the 1980s of MSV resistant maize genotypes, most of the maize genotypes that are currently in use throughout Africa have no, or only low levels of, MSV resistance (*Bosque-Pérez, 2000*; *Martin and Shepherd, 2009*).

For every virus isolate and maize genotype combination, between 36 and 72 three-day old maize seedlings were inoculated in three to five independent experiments using *R radiobacter* mediated delivery of infectious constructs as described by *Martin et al. (2001)*. In each experiment between 12 and 16 plants were inoculated with only *R. radiobacter* growth medium to serve both as leaf size standards relative to which leaf stunting in infected plants would be calculated, and as uninfected cross-contamination controls. Inoculated plants were each grown for 35 days with 16 hr of light per day in approximately 150 grams of soil at between 20 and 25 degrees centigrade in a biosafety level one plant growth room. Leaves four, five and six from each maize seedling were harvested at 21, 28 and 35 days post-inoculation, respectively. The second quarter from the base of each leaf was used for automated symptom quantification by image analysis. Chlorotic leaf area measurements of this leaf segment using image analysis have previously been shown to correlate with a five-point scale widely used by breeders for visually rating the severity of MSV symptoms when assessing the MSV resistance of maize genotypes in the field (*Bosque-Pérez, 2000*; *Martin et al., 1999*).

Leaf segments were scanned at 300 dpi on a HP Scanjet 559OP flatbed scanner (Hewlett Packard, USA), and Windows 32 bit bitmap images were analysed with the custom-written image analysis program, DIA (*Martin, 2019*). All leaf images were captured against a blue background with a circular size standard (a South African 50c coin). For each analysed image, the program DIA (*Martin, 2019*) initially identified and counted the number of pixels in the size standard and then identified the leaf segments as the remaining 'non-blue' objects in the image that were greater than 500 pixels in size. For each identified leaf segment DIA quantified four different symptom phenotypes:

1. The area of the leaf segment relative to the size standard. This was achieved by dividing the number of pixels within the leaf segment image by the number of pixels within the size standard image. Given that the coin has a known area it is then straightforward to adjust the unit of leaf area measurement to $mm^2$. For each agroinoculation experiment in a particular maize genotype, the leaf segment areas thus determined were each individually divided by the average leaf segment areas determined for all of the uninfected control plants of the same genotype from that experiment. For each leaf segment this number was then subtracted from one

to yield the final leaf stunting score (such that lower values of this score would mean less stunting and higher values would mean more stunting).

2. The degree of leaf deformation. This was achieved by fitting the smallest possible rectangle around the leaf segment and then determining the proportion of pixels within this rectangle that fell outside the leaf segment.

3. The proportion of the leaf covered by chlorotic lesions. This was achieved by firstly identifying the main pixel colour categories in each leaf image and then determining which of these pixel colour categories represent chlorotic lesions. The main pixel colour categories were determined by randomly sampling 1000 pixels from the image of the leaf segment, calculating the Euclidean distance between the red, green and blue colour scores of every pair of pixels, constructing a UPGMA dendrogram from these scores and, from this dendrogram, identifying the ten main pixel colour categories. Each pixel in the leaf image was then assigned to one of the ten categories based on the Euclidean distance between the pixel red, green and blue colour scores and the mean of the red, green and blue colour scores of the ten pixel categories. Once all pixels were assigned to categories the average red, green and blue colour scores of each of the ten categories were determined. To determine which of the pixel colour categories represented chlorotic lesions, the ten pixel colour categories were partitioned into the two sets that maximized the difference between the average red+blue colour scores of the sets. The set of pixels with the highest average red+blue score was taken as the 'chlorotic lesion set' and the set of pixels with the lower score as the 'non-chlorotic leaf area set'. The proportion of pixels assigned to the chlorotic lesion set divided by the total number of pixels in the leaf segment image was the proportion of the leaf area falling within chlorotic lesions.

4. The average intensity of chlorosis. This was achieved by taking the average of the red, green and blue colour scores of every pixel in the chlorotic lesion set and dividing by 255 (such that pure white = 1 and pure black = 0).

Outlier measures were identified using Tukey's method (**Tukey, 1977**): symptom measures below the 1.5 interquartile range were discarded. Averages of chlorotic area, intensity of chlorosis, leaf deformation and leaf stunting measurements across leaves 4, 5 and 6 for each virus in each host were later used to infer the magnitudes of these symptom types in ancestral MSV-A variants.

## Inference of ancestral MSV sequences

An alignment containing the 59 MSV-A isolates for which symptoms were quantified (hereafter referred to as 'phenotyped' isolates), together with an additional 630 MSV-A sequences and 182 sequences belonging to MSV strains B through K (all of which can be obtained from GenBank), was constructed using MUSCLE with default settings (**Edgar, 2004**) and edited by eye using the suite of editing tools available in IMPALE (**Khoosal and Martin, 2015**). This 871 MSV full genome sequences alignment was screened for recombination using RDP4.46 (**Martin et al., 2015**) in a two-stage process. In an initial screen, default settings were used and all evidence of recombination within MSV strains other than MSV-A was removed from the alignment by identifying recombinationally-derived sequence fragments and removing these by replacing the associated nucleotide characters in the alignment file with the standard 'gap' character, '-''. In the second screen, other than the use of no multiple testing p-value correction and a p-value cut-off of $10^{-6}$, default RDP4.46 settings were again used. This Bonferroni uncorrected p-value cut-off is still reasonably conservative due to the fact that during the ancestral sequence reconstructions we were specifically focused only on the small subset of all detectable recombination events that impacted the inference of the seven ancestral MSV-A sequences that would later be chemically synthesized.

The ancestral sequence inference was carried out using RDP4.46 (**Martin et al., 2015**) and MrBayes 3.2 (**Ronquist et al., 2012**) with recombination being accounted for in two distinct ways:

1. Whenever the most recent common ancestor (MRCA) of a particular group of MSV-A sequences was to be determined, and every member of that group was inferred to possess evidence of one or more ancestral recombination events (i.e. the MRCA likely also contained evidence of that recombination event), the ancestral sequences were inferred with partitions at the identified recombination breakpoint sites, across which tree topologies were free to vary.

2. For all other recombination events (i.e. those with no evidence of being present within the MRCA sequence being inferred), every nucleotide within all recombinationally-derived genome fragments were replaced in the sequences being analysed using '-'' symbols (in this case denoting missing data).

The inference of MRCA sequences was carried out on subsets of between 252 and 346 of the original 871 sequences, as the full 871 sequence alignment could not be processed in a reasonable amount of time (*Supplementary file 4*; alignments in Supplementary Data 1-3). These subsets were selected based on both the phylogenetic relatedness of sequences to the MRCA being inferred, and the degrees of relatedness of sequences to others in the analysed dataset. Specifically, sequences in MSV-A clades that branched within three nodes to the node of the MRCA being inferred were iteratively removed following two simple rules: (1) if two sequences differed by five or fewer sites and neither was one of the 59 phenotyped sequences, then one was randomly discarded; and (2), if a non-phenotyped sequence differed at five or fewer sites to a phenotyped sequence, then the non-phenotyped sequence was discarded. For those MSV-A sequences falling within clades that branched more than three nodes from the MRCA sequence being inferred, this same iterative sequence removal approach was used except that the rules were applied to sequences differing at ten or fewer sites, instead of five or fewer sites. For the MSV sequences belonging to strains other than MSV-A, sequences were iteratively removed until there were no pairs of non-MSV-A sequences that differed from one another at 20 or fewer sites.

The MRCA sequences that were selected for inference were (i) the MRCA of all the currently sampled MSV-A sequences (designated A0), (ii) the MRCA of all the MSV-A sequences sampled from mainland Africa (designated A1), (iii) the MRCA of all currently sampled members of the MSV-$A_1$ subtype which is currently the most widely distributed MSV-A lineage found in Africa and which is currently the most prevalent subtype found in all parts of Africa other than southern Africa (designated A2), (iv) the MRCA of all currently sampled members of the MSV-$A_3$ subtype which is the second most prevalent MSV-A subtype found in East Africa to date (designated A3), (v) the MRCA of all members of the MSV-$A_4$ subtype which is the most prevalent MSV-A subtype found in southern Africa (designated A4), (vi) the MRCA of all currently sampled members of the second most prevalent MSV-$A_1$ sub-lineage in Africa and the most prevalent MSV-$A_1$ sub-lineage found on Madagascar (designated A5), and (vii) the MRCA of all currently sampled members of the most prevalent MSV-$A_1$ sublineage in Africa (designated A6).

Each MRCA sequence was inferred with MrBayes3.2 using either two or three independent runs, with sampling carried out every 500 generations and all but the last 1000 samples discarded as burn-in. Convergence was monitored using standard deviations of split frequencies, ranging between 0.064 and 0.007 after between 12.2 million and 44.6 million generations. Each MRCA sequence was determined based on the average posterior probabilities of the nucleotide states at each genome site that were yielded by the two/three independent runs (*Supplementary file 4*). At nucleotide sites where no single nucleotide state had an associated posterior probability >0.8 in any of the replicated runs (such as occurred at sites likely to be gap characters), the MRCA sequence inferred by maximum parsimony using PHYLIP (*Felsenstein, 1993*) from a maximum likelihood tree constructed using RAxML8 (*Stamatakis, 2014*) was used either as a tie-breaker, where it was likely that a site should have an associated nucleotide, or to infer that a site should have no associated nucleotide, reflecting what should be a gap character in the MRCA sequence within the context of the analysed alignment. Ancestral genome sequences (Supplementary Data 4) were synthesized at Epoch Life Science (USA). Infectious clones of these reconstructed genomes were produced as in *Martin et al. (2001)*, and were used to infect the S, M and R maize as described above.

## Phylogenetic inference of past virus infection symptoms

Prior to fitting a dated-tip molecular clock model in a Bayesian inference approach, we evaluated the temporal signal using regressions of root-to-tip genetic distances against sequence sampling times. The analyses were based on maximum likelihood trees inferred with the program FastTree 2 (*Price et al., 2010*) and the determination coefficients ($R^2$) of the linear regression were estimated with the program TempEst (*Rambaut et al., 2016*). The p-values were calculated using the approach of *Murray et al. (2016)* and based on 1000 random permutations of the sequence sampling dates (*Navascués et al., 2010*). This root-to-tip regression analysis confirmed the presence of a significant temporal signal ($R^2$ = 0.196, p-value<0.001).

Phylogenetic inference was performed with BEAST 1.10 (*Suchard et al., 2018*) using a combination of flexible models: a relaxed (uncorrelated lognormal) molecular clock with dated tips (*Rambaut, 2000*) and, for the continuous diffusion analysis (i.e. continuous character mapping) of

symptom measurements, a relaxed random walk diffusion model with an underlying lognormal distribution to represent among-branch heterogeneity (*Lemey et al., 2010*). In both cases, posterior estimates for the standard deviations of the lognormal distributions, from which the branch-specific evolutionary rates and the rate scalers for the branch-specific precision matrices are drawn, indicated a significant deviation from a strict molecular clock and a homogeneous Brownian diffusion model. We further opted for a simple constant size coalescent model as tree prior to avoid mixing and convergence problems associated with high-dimensional model parameterisations. We analysed an alignment of 60 sequences from which recombinant regions were removed. The alignment – containing the 59 phenotyped MSV-A isolates and a single phenotyped MSV-B isolate used as an outgroup (alignment in Supplementary Data 4) - was constructed as above using MUSCLE and edited using IMPALE. During phylogenetic inference, clades descending from inferred ancestral sequences were constrained in order to ensure that the node corresponding to the seven reconstructed ancestral sequences existed within the final trees. We ran an MCMC of 500 million generations, sampling every 0.1 million generations and removing the first 2% of samples as burn-in. A maximum clade credibility tree (MCC tree) was then inferred from the posterior tree distribution using TreeAnnotator 1.10 from the BEAST package (*Suchard et al., 2018*). For the comparison, phylogenetic inference of past virus infection symptoms was performed with and without the outgroup (MSV-B) sequence and associated symptom measures. Including or excluding this outgroup in the analysis had no noticeable effect on the ancestral reconstruction of symptom intensities (results not shown).

Finally, we displayed the results of the continuous diffusion analyses using two different approaches: (1) by colouring MCC trees according to inferred symptom values, and (2) by generating figures reporting averaged inferred symptom values associated with phylogenetic branches through time. Note that while this later approach allows estimating general evolutionary trends in symptom intensities through time, it is informed by the number of branches occurring at each time slice. Therefore, when going back in time, the number of branches available to estimate these global trends decreases, and they are increasingly distant for the tip measurements informing these estimates, which reduces the power to infer any deviation from apparent symptom stability. To avoid a strong contribution to inferred symptom intensities by the MSV-B outgroup isolate, these summaries were produced for the phylogenetic analysis performed without including the MSV-B outgroup.

We investigated the phylogenetic signal associated with symptom measurements by estimating Pagel's λ values (*Pagel, 1999*) for every combination of symptom and host type. Pagel's λ is a commonly used measure of phylogenetic signal in continuous traits such as the symptom measurements examined here. Whereas a Pagel's λ value of 0 reflects independence across trait observations and therefore an absence of phylogenetic signal, a value of 1 suggests that traits arise according to a classic Brownian diffusion model in a manner that is perfectly consistent with the phylogenetic relationships of the organisms from which the trait measurements were taken.

## Data availability

All data and R code used for analyses in this study are available on the following public repository: https://github.com/sdellicour/msv_symptom_evolution (*Dellicour, 2020*; copy archived at https://github.com/elifesciences-publications/msv_symptom_evolution).

## Acknowledgements

ALM was supported by a Swedish Institute scholarship (00448/2014) to conduct this research at The Swedish University of Agricultural Sciences (Uppsala), and is currently supported by the Research Council of Norway (267978/E40). SD is supported by the Fonds National de la Recherche Scientifique (FNRS, Belgium) and was previously funded by the Fonds Wetenschappelijk Onderzoek (FWO, Belgium). KAO is supported by the South African National Research Foundation (NRF) and The World Academy of Science (TWAS). PL1 is supported by the European Union: European Regional Development Fund (ERDF), by the Conseil Régional de La Réunion and by the Centre de coopération Internationale en Recherche Agronomique pour le Développement (CIRAD). GWH is supported by a South African National Research Foundation Grant number TTK1207122745. PR was supported by EU grant FP7-PEOPLE-2013-IOF (N° PIOF-GA-2013–622571). PL2 is supported by the European Research Council under the European Union's Horizon 2020 research and innovation programme

(grant agreement no. 725422-ReservoirDOCS), and the Research Foundation Flanders (Fonds voor Wetenschappelijk Onderzoek Vlaanderen, G066215N, G0D5117N and G0B9317N).

## Additional information

### Funding

| Funder | Grant reference number | Author |
|---|---|---|
| Svenska Institutet | 00448/2014 | Adérito L Monjane |
| Fonds De La Recherche Scientifique - FNRS | | Simon Dellicour |
| Fonds Wetenschappelijk Onderzoek | | Simon Dellicour |
| South African National Research Foundation | | Kehinde A Oyeniran |
| The World Academy of Sciences | | Kehinde A Oyeniran |
| European Regional Development Fund | | Pierre Lefeuvre |
| Conseil Régional de La Réunion | | Pierre Lefeuvre |
| Centre de Coopération Internationale en Recherche Agronomique pour le Développement | | Pierre Lefeuvre |
| South African National Research Foundation | TTK1207122745 | Gordon W Harkins |
| European Union Seventh Framework Programme | PIOF-GA-2013-622571 | Philippe Roumagnac |
| European Research Council | 725422-ReservoirDOCS | Philippe Lemey |
| Fonds Wetenschappelijk Onderzoek | G066215N | Philippe Lemey |
| Fonds Wetenschappelijk Onderzoek | G0D5117N | Philippe Lemey |
| Fonds Wetenschappelijk Onderzoek | G0B9317N | Philippe Lemey |
| Research Council of Norway | 267978/E40 | Adérito L Monjane |

The funders had no role in study design, data collection and interpretation, or the decision to submit the work for publication.

### Author contributions

Adérito L Monjane, Conceptualization, Data curation, Formal analysis, Supervision, Validation, Investigation, Methodology; Simon Dellicour, Resources, Software, Formal analysis, Supervision, Validation, Investigation, Visualization, Methodology; Penelope Hartnady, Formal analysis, Investigation, Visualization; Kehinde A Oyeniran, Betty E Owor, Formal analysis, Investigation; Marion Bezuidenhout, Elham Yazdkhasti, Investigation, Writing - review and editing; Daphné Linderme, Philippe Roumagnac, Funding acquisition, Investigation; Rizwan A Syed, Data curation, Investigation; Lara Donaldson, Supervision, Investigation; Shane Murray, Resources, Supervision; Edward P Rybicki, Conceptualization, Resources, Funding acquisition; Anders Kvarnheden, Supervision, Funding acquisition; Pierre Lefeuvre, Investigation, Visualization; Rémy Froissart, Conceptualization, Investigation; Dionne N Shepherd, Conceptualization, Resources, Supervision, Funding acquisition, Investigation, Project administration; Gordon W Harkins, Conceptualization, Formal analysis, Supervision, Funding acquisition, Methodology, Project administration; Marc A Suchard, Resources, Software, Methodology; Philippe Lemey, Conceptualization, Resources, Software, Formal analysis, Supervision,

Investigation, Methodology; Arvind Varsani, Data curation, Supervision, Funding acquisition, Investigation, Methodology, Project administration; Darren P Martin, Conceptualization, Resources, Data curation, Software, Formal analysis, Supervision, Funding acquisition, Validation, Investigation, Visualization, Methodology, Project administration

**Author ORCIDs**
Simon Dellicour https://orcid.org/0000-0001-9558-1052
Rémy Froissart http://orcid.org/0000-0001-8234-1308
Arvind Varsani http://orcid.org/0000-0003-4111-2415
Darren P Martin https://orcid.org/0000-0002-8785-0870

**Decision letter and Author response**
Decision letter https://doi.org/10.7554/eLife.51984.sa1
Author response https://doi.org/10.7554/eLife.51984.sa2

## Additional files

**Supplementary files**
• Supplementary file 1. Regression analysis of mean inferred vs observed symptom trait values.

• Supplementary file 2. Sites within ancestral sequences that could not be determined with a high degree of confidence (sites with a posterior probability of <0.8 in all attempted ancestral sequence inferences).

• Supplementary file 3. MSV isolates for which symptoms were quantified.

• Supplementary file 4. Dataset properties, numbers of generations used and standard deviations (SD) of split frequencies achieved during Bayesian inference of ancestral sequences.

• Transparent reporting form

**Data availability**

All data and R code used for analyses in this study are available on the following public repository: https://github.com/sdellicour/msv_symptom_evolution (copy archived at https://github.com/elifesciences-publications/msv_symptom_evolution).

The following dataset was generated:

| Author(s) | Year | Dataset title | Dataset URL | Database and Identifier |
|---|---|---|---|---|
| Monjane AL, Dellicour S, Hartnady P, Oyeniran KA, Owor BE, Bezeidenhout M, Linderme D, Syed RA, Donaldson L, Murray S, Rybicki EP, Kvarnheden A, Yazdkhasti E, Lefeuvre P, Froissart R, Roumagnac P, Shepherd DN, Harkins GW, Suchard MA, Lemey P, Varsani A, Martin DP | 2020 | Data from: Symptom evolution following the emergence of maize streak virus | https://github.com/sdellicour/msv_symptom_evolution | GitHub, msv_symptom_evolution |

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
