## [Decision Letter]

**Acceptance summary:**

The study by Monjane and colleagues examines the changes of four infection-related symptoms of maize streak virus (MSV) since the virus' emergence ~110 years ago. By reconstructing the phylogeny of representative MSV isolates, the authors traced the evolution of symptom intensities across the phylogeny and showed that whereas harm to the host inflicted by the pathogen remained constant or decreased over time, colonization of host cells (on which the insect vectors responsible for MSV transmission feed) increased over time, suggesting the existence of a trade-off. Reconstruction of ancestral genotypes and testing of their inferred and actual symptom intensities was consistent with the existence of trade-offs. This work is novel in two ways. First, it takes advantage of a robust phylogenetic framework to test a major hypothesis in (plant) virus evolution (the existence and extent of fitness trade-offs). Second, it is able to demonstrate that trade-offs also exist in pathogens that exhibit broad host ranges and do not kill their hosts, both characteristics atypical for standard models of the evolution of virulence. With respect to future work, it would be very interesting to examine the results of this study with those of future studies that employ experimental evolution approaches to examine how a pathogen's symptoms related to virus transmission and/or harm to the host could evolve.

**Decision letter after peer review:**

Thank you for submitting your article "Symptom evolution following the emergence of maize streak virus" for consideration by *eLife*. Your article has been reviewed by three peer reviewers, one of whom is a member of our Board of Reviewing Editors, and the evaluation has been overseen by Ian Baldwin as the Senior Editor. The reviewers have opted to remain anonymous.

The reviewers have discussed the reviews with one another and the Reviewing Editor has drafted this decision to help you prepare a revised submission.

Summary:

The study by Monjane and colleagues examines the changes of four infection-related symptoms of maize streak virus (MSV) since the virus' emergence ~110 years ago. By reconstructing the phylogeny of representative MSV isolates, the authors traced the evolution of symptom intensities across the phylogeny and showed that whereas harm to the host inflicted by the pathogen remained constant or decreased over time, colonization of host cells (on which the insect vectors responsible for MSV transmission feed) increased over time, suggesting the existence of a trade-off. Reconstruction of ancestral genotypes and testing of their inferred and actual symptom intensities was consistent with the existence of trade-offs. Therefore, one key conclusion of this study is that trade-offs also exist in pathogens that exhibit broad host ranges and do not kill their hosts, both characteristics atypical for standard models of the evolution of virulence. Thus, a major advantage of this work is the original application of a phylogenetic framework to test a major hypothesis in (plant) virus evolution (trade-off). The difficulty results from the complexity of the biological system studied and its temporal (a century) and implicitly spatial (a continent, Africa) scales. The uncertainties at key steps of the process (some acknowledged and examined by the authors) make difficult the extraction of a level of information adequate to reach the objectives despite the available knowledge, the data acquired, and the range of statistical and phylogenetic analyses implemented. Additional information and complementary analyses may strengthen the conclusions.

Essential revisions:

1) How were the 59 MSV-A isolates selected? Everything the authors infer about the four infection symptom types is conditioned on these isolates being representative of MSV-A diversity. The authors state in the Materials and methods that these isolates are representative but no (phylogenetic or other) evidence is offered showing: a) that this is indeed the case with respect to geography, time, etc. or b) the criteria the authors used to pick "representative" isolates.

2) It is not clear why the authors chose to infer the evolution of symptoms using a phylogenetic approach (instead of directly sampling and analyzing viruses isolated in the last ~110 years). Please explain and justify.

3) Paragraph six of subsection “Increase of MSV-induced chlorotic leaf areas over the past ~110 years of MSV evolution”: The fact that 10% of the isolates examined may not be representative of the virus infecting a given plant seems to be huge caveat that should be taken into account not just in the interpretation of Figure 1, but in the interpretation of all results discussed by this study. We suggest that the authors turn this paragraph into a separate section where they discuss the implications of this sampling issue for the results shown in Figures 1-3.

4) The end of the manuscript is abrupt. What is lacking is a concluding statement summarizing the implications of this work for broad-host-range pathogens. How does this work extend or revise the paradigm?

5) It was not clear how older viral sequences were treated in phylogenetic analyses (e.g., a strain from 1979 could be the ancestor of several strains sampled in later years). Was this taken into account?

6) Also not clear how the time calibration of the phylogeny was performed. Was the origin of the phylogeny set to 110 years ago? How can the authors be sure the origin of MSV-A was 110 years ago and not say 150 or 200 years ago?

7) The authors should better explain and provide evidence on the selection of models for the inference of ancestral symptom states. For example, they use a relaxed (uncorrelated lognormal) molecular clock, a constant population size and then perform a continuous diffusion analysis. Do these models were properly compared with other available in BEAST? And is there any statistical evidence for the selection of this combination of models and parameters? Choice of models and parameters could affect the general conclusions of this work and it is important that this choice is clearly explained.

8) Figure 2 concerns:

– Why was an outgroup included in the time-rooted phylogenetic tree generated by Beast?

– What would be the impact of similar analyses conducted without an outgroup or with alternative outgroup(s) on (1) the symptoms, (2) the Pagel parameter, (3) the ancestral sequences?

– In particular, what are the results when choosing as an outgroup an isolate inducing a small chlorotic area? Do they match the results obtained with the outgroup selected which induced a large chlorotic area?

– There is no time-scale in Figure 2.

9) Figure 3 concerns:

– We had difficulties to match the results of Figure 3 with those of Figure 2. For instance, Figure 3 A shows an increase of chlorotic area with time whereas Figure 2A (top and middle) shows an overall change of color over time from red to green.

– More generally, we found it difficult to read the green-to-red color scales (especially orange) to illustrate changes in symptom intensity. Increasing intensities of the same color (even grey) may be more readable.

– How to explain the changes in symptom intensity over the last 20 years in Figure 3B, 3C and 3D which contrast with the stability all over the century before? Can it be an artifact?

– Figure 2—figure supplement 3B and D. The trends heavily depend on the first point (year 1900). Can this be an effect of inclusion of the outgroup?

10) Figure 4 concerns: Figure(s) illustrates the results of symptoms observed vs inferred for the seven nodes at three levels of resistance. For each symptom component, the correlation was calculated by merging the results of the three levels of resistance. This can be misleading as the link between observed and inferred results should be tested for a given level of resistance. The case of Figure 4 D "leaf stunting", – where the overall correlation is the highest, – is illustrative. At each level of resistance, the inferred symptom (i.e. leaf stunting) is similar whereas the observed symptoms are quite variable. Accordingly, there is no link between inferred and observed leaf stunting. So the correlation found only reflected the levels of resistance of the cultivars: the points at the top right come from the susceptible variety, those at the bottom left from the highly resistant variety, the intermediate ones from the moderately resistant cultivar. In other words, if one selects 7 isolates and randomly associates an isolate to each of them, and reports as points the symptoms of each of the 7 couples (randomly designed), an overall correlation would still be found for the three varieties confounded. This possible bias, – quite apparent for "leaf stunting", – applies to some extent to the other symptom components, with the possible exception of "chlorotic area" on the susceptible cultivar.

– We suggest testing the relationships between inferred and observed symptoms for each given level of resistance/susceptibility. The level of significance should be assessed by methods taking into account the phylogenetic dependence between the isolates such as tip permutation.

– There is a mistake in Figure 4A: there are 6 points instead of 7 for the moderately resistant cultivar and 8 instead of 7 for the susceptible cultivar.

11) Are the conclusions dependent of the choice of the 7 nodes? What confidence is there that similar conclusions would be reached with a different selection of nodes?

---

## [Author Response]

Essential revisions:1) How were the 59 MSV-A isolates selected? Everything the authors infer about the four infection symptom types is conditioned on these isolates being representative of MSV-A diversity. The authors state in the Materials and methods that these isolates are representative but no (phylogenetic or other) evidence is offered showing: a) that this is indeed the case with respect to geography, time, etc. or b) the criteria the authors used to pick "representative" isolates.

Thank you for bringing this to our attention. We have now provided a maximum likelihood tree for all available MSV-A clones (corresponding to more than 700 sequences) including the 59 isolates selected in the present study (see the new Figure 2—figure supplement 4). As we now detail in the Materials and methods section, isolates were selected as follows: (i) we made and tested agroinfectious constructs for every virus that was sampled before 1990 and for which we had a molecular clone sampled directly from a source plant, (ii) we obtained and tested all agroinfectious clones that had previously been constructed in the 1990s and 1980s, and (iii) for viruses sampled between 2000 and 2007, we chose isolates that were representative of the major phylogenetic clades of viruses in eastern, western and southern Africa.

2) It is not clear why the authors chose to infer the evolution of symptoms using a phylogenetic approach (instead of directly sampling and analyzing viruses isolated in the last ~110 years). Please explain and justify.

The phylogenetic approach allowed us to reconstruct and investigate the change of symptoms in an evolutionary context and to control for shared ancestry in analysing differences in viral traits of isolated viruses. In this framework, we were for instance able to establish if closely related viruses tended to display similar symptoms. Furthermore, because our phylogenetic estimates indicate that this is generally the case, an approach is needed to account for phylogenetic correlation. Besides, we actually included the oldest available MSV samples in the analysis. Despite searches of plant pathology herbaria in southern Africa, eastern Africa, France and the United Kingdom, as well as the diseased plant leaf sample collections from every currently functioning lab that has published on MSV over the past 40 years, we were unable to find any samples dating to before 1979. In that context, the phylogenetic and related character mapping approaches constitute relevant solutions to estimate the symptoms displayed by plants infected by ancestral lineages. The non-availability of ancestral sequences is now explicitly stated in the text as a motivation for selecting a phylogenetic approach allowing evolutionary investigations.

3) Paragraph six of subsection “Increase of MSV-induced chlorotic leaf areas over the past ~110 years of MSV evolution”: The fact that 10% of the isolates examined may not be representative of the virus infecting a given plant seems to be huge caveat that should be taken into account not just in the interpretation of Figure 1, but in the interpretation of all results discussed by this study. We suggest that the authors turn this paragraph into a separate section where they discuss the implications of this sampling issue for the results shown in Figures 1-3.

As requested, the impact of isolate sampling is now discussed in a distinct subsection. The reviewers are right when stating that, even if only impacting approximately 10% of the isolates, we cannot exclude that this artefactual correlation could also, to some extent, impact the continuous character mapping of symptom intensities and inferred changes in overall symptom intensities. However, because these 10% of isolates are a priori randomly distributed in the tree, their inclusion is mainly expected to add some random noise in the comparative analysis. Therefore, we do not expect a systematic bias in generating the trends we estimate and compare.

4) The end of the manuscript is abrupt. What is lacking is a concluding statement summarizing the implications of this work for broad-host-range pathogens. How does this work extend or revise the paradigm?

We agree and have now added two new sentences at the end of the Discussion to end with a concluding statement.

“The trade-off we identify is of significant importance because it implies that even when pathogens have host ranges that include over 100 species, within the context of individual host species, they may still conform to evolutionary imperatives that are similar to those of narrow host-range pathogens. Possibly more significant, however, is that a trade-off between sub-lethal symptom severities and transmission rates in MSV alludes to a more general form of the popular transmission-virulence trade-off hypothesis (Alizon et al., 2009; Anderson and May, 1982) which has, until now, focused almost exclusively on the relationship between pathogen-induced host mortality and pathogen transmission rates.”

5) It was not clear how older viral sequences were treated in phylogenetic analyses (e.g., a strain from 1979 could be the ancestor of several strains sampled in later years). Was this taken into account?

The older sequences were not included as common ancestors but as sampled sequences. Indeed, because they are represented by tips in the phylogenetic comparative analyses, these viruses are generally not considered as direct ancestors of more recently sampled viruses. However, an older sample can due to its phylogenetic placement be more closely related to the common ancestor of several recent samples, and an estimated branch length close to zero for such old tip would make it virtually indistinguishable from a sampled ancestor.

6) Also not clear how the time calibration of the phylogeny was performed. Was the origin of the phylogeny set to 110 years ago? How can the authors be sure the origin of MSV-A was 110 years ago and not say 150 or 200 years ago?

Time estimates were obtained by inferring time-calibrated phylogenetic trees using the Bayesian genealogical inference framework in the BEAST software package. For rapidly evolving viruses like MSV, this approach allows exploiting the temporal signal present in the data set, that is the divergence accumulating between samples of successive sampling years, to calibrate a tip-dated molecular clock model (see e.g. Rambaut, 2000; Drummond et al., 2006, doi.org/10. 1371/journal.pbio.0040088). In this model the evolutionary rate is an estimable parameter (equivalent to the slope of divergence through time) that takes care of the rescaling of the tree from units of genetic distance to time. We believe that this aspect was not made sufficiently clear because we did not include the temporal signal analysis prior to fitting a tip-dated clock model in our manuscript. To clarify this issue, we now report the result of this analysis that assesses the presence of sufficient temporal signal for timecalibration in the data set:

“Prior to fitting a dated-tip molecular clock model in a Bayesian inference approach, we evaluated the phylogenetic temporal signal using regressions of root-to-tip genetic distances against sequence sampling times. The analyses were based on maximum likelihood trees inferred with the program FastTree 2 (Price et al., 2010) and the determination coefficients (R2) of the linear regression were estimated with the program TempEst (Rambaut et al., 2016). The p-values were calculated using the approach of Murray et al., 2016, and based on 1,000 random permutations of the sequence sampling dates (Navascuès et al., 2010). This root-to-tip regression analysis confirmed the presence of a significant temporal signal (R2 = 0.196, p-value < 0.001).”

We have also added “with dated tips” when referring to the molecular clock model and an appropriate reference for this model:

“… a relaxed (uncorrelated lognormal) molecular clock with dated tips (Rambaut, 2000) …”

In addition, we now also report the maximum clade credibility (MCC) tree with credible (95% HPD) intervals for internal node age estimates (Figure 2—figure supplement 2). As you can see in the new Figure 2—figure supplement 2, the credible interval for instance estimated for the root age is relatively large as one would expect for nodes that go relative deep into time with respect to the sampling time range.

7) The authors should better explain and provide evidence on the selection of models for the inference of ancestral symptom states. For example, they use a relaxed (uncorrelated lognormal) molecular clock, a constant population size and then perform a continuous diffusion analysis. Do these models were properly compared with other available in BEAST? And is there any statistical evidence for the selection of this combination of models and parameters? Choice of models and parameters could affect the general conclusions of this work and it is important that this choice is clearly explained.

We agree that the motivation for different model choices was not made explicit. BEAST analyses fit a full probabilistic model that combines different model components, and for each component different parameterisations are available. While formal model fit measures can be computed in Bayesian phylogenetic inference, these are computationally highly demanding and therefore unpractical for a complete assessment of all combinations of model components. Furthermore, for some model components, they are associated with problems or remain untested. Therefore, we restrict their use for testing models that constitute a hypothesis we are testing, or for testing nuisance models that may impact our estimates and for which no obvious prior choice is available or standard posterior model estimates are not helpful in arguing model choices. For the models that can impact our estimates, the relaxed clock model and the relaxed random walk (RRW) model, we can rely on the posterior estimates to indicate their better fit compared to the simpler strict molecular clock and strict Brownian diffusion model, respectively. Specifically, we can consider the posterior estimate of the standard deviation of the lognormal distribution from which the branch-specific evolutionary rates and the rate scalers for the branch-specific precision matrices ae drawn. A value of zero for this parameter would reflect a strict molecular clock and homogeneous Brownian diffusion process respectively. In both cases however, our mean estimates diverge from 0 with HPD intervals that do not include 0 and therefore reject the simple null models. For instance, the standard deviation of the lognormal distribution associated with the relaxed clock was estimated to be 0.000305 with a 95% HPD interval equal to [0.000123-0.000524]. In this case we opted for a simple constant-size population model for two reasons: (i) it was not our goal to estimate the dynamics of population size change through time and (ii) simpler models help avoiding convergence and mixing problems in Bayesian inference that are particularly problematic for high-dimensional model. This has now also been made explicit:

“Phylogenetic inference was performed with BEAST 1.10 (Suchard et al., 2018) using a combination of flexible models: a relaxed (uncorrelated lognormal) molecular clock with dated tips (Rambaut, 2000) and, for the continuous diffusion analysis (i.e. continuous character mapping) of symptom measurements, a relaxed random walk diffusion model with an underlying lognormal distribution to represent among-branch heterogeneity (Lemey et al., 2010). In both cases, posterior estimates for the standard deviations of the lognormal distributions, from which the branch-specific evolutionary rates and the rate scalers for the branch-specific precision matrices are drawn, indicated a significant deviation from a strict molecular clock and a homogeneous Brownian diffusion model. We further opted for a simple constant size coalescent model as tree prior to avoid mixing and convergence problems associated with high-dimensional model parameterisations.”

8) Figure 2 concerns:– Why was an outgroup included in the time-rooted phylogenetic tree generated by Beast?

While an outgroup is indeed not necessary to root the trees in a BEAST analysis, we included it for two reasons. As we have symptom measurements for the outgroup strain that are often relatively different from the mean ingroup trait, we thought it could help inform the trait of the common ancestor of the ingroup. In addition, the inclusion of the root contributes to a significant phylogenetic signal. Indeed, without the outgroup, our preliminary analysis indicates a non-significant phylogenetic signal, i.e. a p-value just above the significance threshold of 0.05 (p-value = 0.079).

– What would be the impact of similar analyses conducted without an outgroup or with alternative outgroup(s) on (1) the symptoms, (2) the Pagel parameter, (3) the ancestral sequences?

To address this comment, we ran new analyses without this outgroup. As now stated in the text, the results of the new analysis confirm that the outgroup has no meaningful impact on Pagel’s λ estimates, on the ancestral reconstruction of symptoms associated to the MSV-A clade, and therefore on the correlation between inferred and observed symptoms associated with ancestral sequences. In Author response table 1, we summarise the comparison between Pagel’s λ estimated in the analysis with and without the outgroup respectively:

**Author response table 1. resptable1:** 

Genotype	Analysis	Chlorotic area	Intensity of chlorosis	Leaf deformation	Leaf shunting
**S**	with outgroup	0.545 [0228-0.847]	0.289 [0.026-0.582]	0.456 [0.113-0.799]	0.608 [0.300-0.900]
	without outgroup	0.547 [0.228-0.852]	0.293 [0.029-0.590]	0.460 [0.117-0.803]	0.609 [0.288-0.895]
**M**	with outgroup	0.503 [0.185-0.818]	0.658 [0.347-0.928]	0.217 [0.006-0.506]	0.549 [0.214-0.870]
	without outgroup	0.501 [0.182-0.805]	0.661 [0.352-0.927]	0.215 [0.006-0.492]	0.549 [0.222-0.880]
**R**	with outgroup	0.561 [0.216-0.892]	0.428 [0.138-0.718]	0.400 [0.031-0.782]	0.403 [0.042-0.774]
	without outgroup	0.557 [0.214-0.886]	0.426 [0.141-0.727]	0.399 [0.036-0.790]	0.405 [0.057-0.783]

In Author response table 2 we also summarise the comparison of the ancestral symptom values estimated for the most recent common ancestor of MSV-A selected in the present study (i.e. ancestor A0):

**Author response table 2. resptable2:** 

Genotype	Analysis	Chlorotic area	Intensity of chlorosis	Leaf deformation	Leaf shunting
**S**	with outgroup	46.9 [35.9, 58.0]	0.604 [0.578, 0.623]	2.459 [2.064, 2.950]	0.784 [0.629, 0.872]
	without outgroup	50.4 [40.3, 60.3]	0.604 [0.579, 0.627]	2.512 [2.038, 3.189]	0.800 [0.636, 0.887]
**M**	with outgroup	37.9 [21.3, 54.9]	0.557 [0.492, 0.621]	1.789 [1.616, 1.968]	0.526 [0.440, 0.603]
	without outgroup	42.3 [27.0, 58.1]	0.558 [0.485, 0.628]	1.801 [1.620, 2.018]	0.522 [0.421, 0.608]
**R**	with outgroup	24.3 [12.3, 38.3]	0.573 [0.525, 0.625]	1.640 [1.449, 1.934]	0.362 [0.240, 0.449]
	without outgroup	26.4 [14.7, 40.9]	0.573 [0.519, 0.627]	1.732 [1.433, 2.100]	0.386 [0.280, 0.462]

Finally, in Author response image 1 we provide a comparison between correlation values estimated from the phylogenetic analyses performed while including or excluding the outgroup:

– In particular, what are the results when choosing as an outgroup an isolate inducing a small chlorotic area? Do they match the results obtained with the outgroup selected which induced a large chlorotic area?

Unfortunately, we do not have the measures associated with such an alternative outgroup. However, as shown above, the inclusion of an outgroup had almost no impact on the inference of past symptom values within the MSV-A clade. It therefore seems safe to conclude that an alternative outgroup would not have changed the overall evolutionary picture that we report.

– There is no time-scale in Figure 2.

We have now added a time-scale.

9) Figure 3 concerns:– We had difficulties to match the results of Figure 3 with those of Figure 2. For instance, Figure 3 A shows an increase of chlorotic area with time whereas Figure 2A (top and middle) shows an overall change of color over time from red to green.

We apologise for this mistake in the previous version of Figure 2A. The colour scale was erroneously inverted, and this is now fixed in the updated version of that figure. Therefore, Figure 2A is now fully coherent with Figure 4A (previously Figure 3A).

– More generally, we found it difficult to read the green-to-red color scales (especially orange) to illustrate changes in symptom intensity. Increasing intensities of the same color (even grey) may be more readable.

Following this remark, we have changed the colour scale and now use a grey scale (light grey = weaker symptoms, and dark grey = stronger symptoms).

– How to explain the changes in symptom intensity over the last 20 years in Figure 3B, 3C and 3D which contrast with the stability all over the century before? Can it be an artifact?

This pattern reflects the fact that there is no sufficiently strong information to inform estimates of trait dynamics before the sampling time range. The symptom intensities are averaged over branches occurring in each time slice (and there is an HPD interval as we averaged these values for several trees sampled from the posterior distribution). Therefore, the more we go back in time, the less branches we have to average the overall trend in symptom intensities and the further away these branches are from the observations at the tips. This aspect is now formally stated in the text. We also refer to our answer to the next comment regarding Figure 4 (previously Figure 3).

“Note that while this later approach allows estimating general evolutionary trends in symptom intensities through time, it is informed by the number of branches occurring at each time slice. Therefore, when going back in time, the number of branches available to estimate these global trends decreases, and they are increasingly distant for the tip measurements informing these estimates, which reduces the power to infer any deviation from apparent symptom stability.”

– Figure 2—figure supplement 3B and D. The trends heavily depend on the first point (year 1900). Can this be an effect of inclusion of the outgroup?

Yes, indeed, thank you for pointing this. This aspect is thus also related to the limitation pointed in our answer to the previous comment: the more we go back in time, the less branches we have to estimate the overall symptom trends and the further they are away from the tip observations. The issue is further aggravated by the fact that between 1900 and 1950, the branch connecting the outgroup is one of the few (or even one of the two) branches remaining. To circumvent this issue, we have now re-generated these graphs on the new analysis performed without the outgroup (Figure 4, previously Figure 3).

10) Figure 4 concerns: Figure(s) illustrates the results of symptoms observed vs inferred for the seven nodes at three levels of resistance. For each symptom component, the correlation was calculated by merging the results of the three levels of resistance. This can be misleading as the link between observed and inferred results should be tested for a given level of resistance. The case of Figure 4 D "leaf stunting", where the overall correlation is the highest, is illustrative. At each level of resistance, the inferred symptom (i.e. leaf stunting) is similar whereas the observed symptoms are quite variable. Accordingly, there is no link between inferred and observed leaf stunting. So the correlation found only reflected the levels of resistance of the cultivars: the points at the top right come from the susceptible variety, those at the bottom left from the highly resistant variety, the intermediate ones from the moderately resistant cultivar. In other words, if one selects 7 isolates and randomly associates an isolate to each of them, and reports as points the symptoms of each of the 7 couples (randomly designed), an overall correlation would still be found for the three varieties confounded. This possible bias, quite apparent for "leaf stunting", applies to some extent to the other symptom components, with the possible exception of "chlorotic area" on the susceptible cultivar.– We suggest testing the relationships between inferred and observed symptoms for each given level of resistance/susceptibility. The level of significance should be assessed by methods taking into account the phylogenetic dependence between the isolates such as tip permutation.

The reviewers make a good point and we now acknowledge the absence of within genotype correlation for the mean estimates. This is interesting in its own right as it shows that the models and inference approaches would benefit from formally including the ancestral measurements, which is a motivation for us to further extend our phylogenetic comparative framework. Controlling for shared ancestry in estimating/testing such correlations is a good suggestion, but as our correlations within genotypes are low/non-significant in the first place, there is no need to do so in the current situation. Not controlling for shared ancestry could lead to overconfidence in correlations or induce false positives in the testing, but we do not estimate high correlations, nor do we see significance in the present case.

– There is a mistake in Figure 4A: there are 6 points instead of 7 for the moderately resistant cultivar and 8 instead of 7 for the susceptible cultivar.

Indeed, thank you for pointing this out – this mistake has now been fixed.

11) Are the conclusions dependent of the choice of the 7 nodes? What confidence is there that similar conclusions would be reached with a different selection of nodes?

While we do not have the data to prove this, our expectation is that our results should be robust regarding the choice of these 7 nodes as long as they represent a similar diversity of lineages. Furthermore, these 7 nodes were not selected at random. Indeed, as now more clearly described in the text, these ancestral sequences are the MRCA sequences of the main MSV-A clades and/or represent epidemiologically important progenitors of the main currently circulating MSV-A lineages. We hope that our study demonstrates the potential of using ancestral viral measurements in evolutionary studies, which could stimulate applications with many more node measurements making a particular selection less relevant.